

# Locating trees to mitigate outdoor radiant load of humans in urban areas using a metaheuristic hill climbing algorithm – Introducing TreePlanter v1.0

Nils Wallenberg[1], Fredrik Lindberg[1], David Rayner[1]

[1]Department of Earth Sciences, University of Gothenburg, Gothenburg, 413 20, Sweden

*Correspondence to*: Nils Wallenberg (nils.wallenberg@gvc.gu.se)

**Abstract.** Mean radiant temperature ($T_{mrt}$) is a frequently-used measure of outdoor radiant heat conditions. Excessive $T_{mrt}$, linked especially to clear and warm days, have negative effect on human well-being. Highest $T_{mrt}$ on such days is found in sunlit areas, whereas shaded areas have significantly lower values. One way of alleviating high $T_{mrt}$ is by planting trees to provide shade in exposed areas. To achieve the most efficient mitigation of excessive $T_{mrt}$ by tree shade with multiple trees requires optimized positioning of the trees, which is a computationally extensive procedure. By utilizing metaheuristics, calculations can be reduced. Here, we present TreePlanter v1.0, which applies a metaheuristic hill climbing algorithm on input raster data of $T_{mrt}$ and shadow patterns to position trees in complex urban areas. The hill climbing algorithm enables dynamic exploration of the input data to position trees, compared with very computationally demanding brute-force calculations. The results show that the algorithm, in relatively low model runtime, can find positions for several trees simultaneously that lowers $T_{mrt}$ substantially. TreePlanter can assist in future research on optimization of tree planting in urban areas to increase thermal comfort. The current version can only position trees of equal tree characteristics (tree height, tree canopy and trunk height). Expected developments include positioning of trees with different tree characteristics.

## 1 Introduction

The increased risk of exposure to excessive heat in urban areas during extreme events as an effect of a modified urban climate can lead to excess mortality and morbidity (Dousset et al., 2010; Gabriel and Endlicher, 2010). The modified and generally warmer urban climate is a result of several factors, such as density of building stock, street orientation, color of materials, absence of permeable surfaces, anthropogenic heat and lack of vegetation (Arnfield, 2003). Mean radiant temperature ($T_{mrt}$ (°C)) is an important meteorological variable in the human energy balance and outdoor human thermal comfort especially during clear and warm weather (Mayer and Höppe, 1987; Höppe, 1992; Mayer et al., 2008). $T_{mrt}$ is essentially the sum of all short- and longwave radiation fluxes (both direct and reflected) to which the human body is exposed, defined by ASHRAE (2001) as the "uniform temperature of an imaginary enclosure in which radiant heat transfer from the human body equals the radiant heat transfer in the actual non-uniform enclosure". High $T_{mrt}$ have negative effects on human health (Mayer et al., 2008). Thorsson et al. (2014), for example, showed that there is a higher correlation between $T_{mrt}$ and mortality compared to





air temperature and mortality on hot days. High daytime $T_{mrt}$ correlated with heat related mortality among people aged 80+, whereas high nighttime $T_{mrt}$ correlated with heat related mortality among peopled aged 45-79.

Lateral longwave irradiance surroundings is the largest component of $T_{mrt}$ (Lindberg et al., 2014), and although this component does increase on clear days, high values of outdoor $T_{mrt}$ only occur in sunlit areas exposed to direct shortwave irradiance from the sun (Lindberg and Grimmond, 2011a). $T_{mrt}$ peaks around noon and early afternoon when sun is at its highest position

during the day. Areas in front of sunlit south facing walls (northern hemisphere) are exposed to high radiant load (Lindberg et al., 2016; Wallenberg et al., 2020), due to high amount of shortwave irradiance from the sun, but also reflected shortwave irradiance from adjacent walls, as well as emitted longwave irradiance from nearby warm surfaces (Lindberg et al., 2016; Thorsson et al., 2017). A proven method to tackle the issue of high daytime $T_{mrt}$ on clear days is through shading by e.g. buildings or vegetation (Lindberg and Grimmond, 2011b; Srivanit and Jareemit, 2020). Buildings are static and their

geometries challenging to modify, whereas increasing the fraction of vegetation is a favored heat mitigating strategy (Konarska et al. 2014; Norton et al., 2015; Lindberg et al., 2016).

Reducing solar radiation (e.g. Bajsanski et al., 2016; Stojakovic et al., 2020) and $T_{mrt}$ (e.g. Konarska et al., 2014; Lindberg et al., 2016; Zhao et al., 2018; Abdi et al., 2020; Lee et al., 2020; Lee and Mayer, 2020; HosseiniHaghighi et al., 2020) with tree shade has been extensively studied. Studies on the effects of vegetation on thermal comfort often focus on either vegetation in

general, or more specifically on the positioning and arrangement of the trees. Zhao et al. (2018) studied the effects of trees on a neighborhood in Phoenix (Texas, US) and showed that two trees with equal distance had a higher effect on thermal comfort compared to two dispersed trees. Lee et al. (2020) analyzed the distance between trees in combination with tree canopy size and showed that the shading effect of trees increases with a lower aspect ratio (height/width ratio, H/W) of surrounding buildings. They also showed that a larger tree canopy in combination with an increased distance between the trees had a larger

positive effect on thermal comfort on the northern sidewalk in an east-west oriented street canyon, compared to the southern sidewalk. This effect was mainly attributable to a decrease in $T_{mrt}$. Trees had, on the other hand, little effect on air temperature regardless of whether the position was sunlit or shaded.

A common denominator within the topic of trees as a mitigating tool, regardless of whether the focus is on distance between trees or tree size, is that the tree or trees are located without knowledge of how optimally they are placed with respect to

reducing radiant load. Even though the locations of the tree or trees have an effect on the thermal comfort (Abdi et al., 2020; Lee et al., 2020; Srivanit and Jareemit, 2020), optimized positioning could enhance the mitigating effect. One way of addressing this issue is to calculate the optimal positions of the trees using spatial information (insolation and shadow patterns), and the possible locations for the trees (i.e. where there are no buildings or other obstructing structures). However, brute-force calculations for optimal positions quickly become extremely computationally demanding with several trees. A different, less

demanding method of finding optimal locations for trees is by exploiting metaheuristic algorithms. Examples of metaheuristic algorithms are the genetic or evolutionary algorithms. Such algorithms inherit "genes", e.g. coordinates of positions of trees, from previous "populations", e.g. an iteration in an algorithm for positioning the trees. Stojakovic et al. (2020) utilized an evolutionary algorithm for positioning trees to mitigate insolation in a rectangular city block in Belgrade, Serbia, and clearly



showed that locations differ depending on prerequisites like height of surrounding buildings. Chen et al. (2008) and Ooka et

al. (2008) exploited genetic algorithms to study optimal positions for trees and tree arrangements using computational fluid

dynamics simulations in a hypothetical urban block in Tokyo. Another example of a metaheuristic algorithm is the greedy

algorithm, utilized by Zhao et al. (2017), to optimize tree locations to study shading effect and shade coverage on building

facades. This algorithm finds the optimal position for one tree at a time, which means that when the position for the first tree

is established it is occupied, and cannot be adjusted. A hill climbing algorithm (Luke, 2013) is an additional example, where

neighboring positions of a tree are explored to identify a better position concerning e.g. shading and reduction of radiant load.

Here we present TreePlanter, a model for optimization of tree positions to mitigate heat stress by reducing outdoor radiant load

in urban settings. The optimization of tree positions is achieved by utilizing a metaheuristic hill climbing algorithm to reduce

$T_{mrt}$. TreePlanter is incorporated in the Urban Multi-Scale Environmental Predictor climate service tool (UMEP; Lindberg et

al., 2018), to facilitate usage by other researchers, and practitioners such as urban planners and landscape architects

([http://umep-docs.readthedocs.io/](http://umep-docs.readthedocs.io/)).

## 2 Methods

TreePlanter is built upon output data generated by the SOlar and LongWave Environmental Irradiance Geometry (SOLWEIG)

model (Lindberg et al., 2008). SOLWEIG is a 2.5D-model able to estimate spatial variations of $T_{mrt}$ using commonly-available

meteorological forcing data (incoming shortwave radiation, air temperature and relative humidity) and surface information

such as a digital surface model (DSM) including the elevation of buildings and ground (e.g. Fig. 3b). Developments in

SOLWEIG enable inclusion of 3D vegetation data (Lindberg and Grimmond, 2011b) and variations in ground surface cover

(Lindberg et al., 2016). SOLWEIG has been subject of evaluation in several studies (e.g. Lindberg and Grimmond, 2011b;

Lindberg et al., 2016; Lau et al., 2016; Chen et al., 2016; Kántor et al. 2018; Gál and Kantor, 2020). Furthermore, SOLWEIG

is a popular radiation model, utilized extensively in applied studies (e.g. Lindberg et al., 2013; Lau et al., 2014; Jänicke et al.,

2015; Thom et al., 2016; HosseiniHaghighi et al., 2020).

TreePlanter uses the gridded $T_{mrt}$ and shadow patterns output from SOLWEIG, as well as information on the locations of

buildings, and the meteorological forcing data used by SOLWEIG. Example outputs from SOLWEIG are shown in Fig. 3c

and 3d. The figures show output of $T_{mrt}$ and shadow patterns, respectively, for 1700 LST on June 22 1983. By comparing $T_{mrt}$

with shadow patterns, it is evident that radiant load in shaded areas is substantially lower compared to sunlit areas. Output data

from SOLWEIG is crucial for running the particular model described here. However, in theory, any raster data of $T_{mrt}$ and

shadow patterns could be used.

A planting area (study area) is defined inside the spatial extent of the output data from SOLWEIG. This planting area can

cover either the entire spatial extent of the output data, or be delimited to a smaller area, to confine positioning of trees to areas

within the model domain that are not occupied by buildings or existing trees. Furthermore, it is not possible to position trees

within one radian of the tree canopy diameter from buildings or other trees. See Sect. 5.1 for further description.





## 2.1 Model pre-processing

A general and simplified flowchart of the model pre-processing is shown in Fig. 1a. First, a general 3D tree form is designed from input tree morphology data: tree type (deciduous or conifer), tree height, trunk zone height (height from ground to canopy, i.e. bare trunk) and canopy diameter. The meteorological forcing data (same as for the SOLWEIG run) is used to estimate $T_{mrt}$

for a point that is shaded by a tree with a transmissivity ($\tau$) to shortwave radiation for specified time steps (specified by the meteorological data). Additionally, the specified time steps from the meteorological data are used to generate corresponding shadows in a fictitious flat environment (i.e. no buildings or trees), using the general 3D tree shape produced from the input tree morphology data (hereafter referred to as $shadows_{tree}$).

With the generated $shadows_{tree}$ and calculated $T_{mrt}$ for a point shaded by a tree for each time step, $T_{mrt}$ in the tree shadows for

each time step can be estimated (hereafter referred to as $T_{mrt.tree}$). By iterating over every position in the planting area where a tree can potentially be located and moving only one tree, it is possible to estimate the difference in $T_{mrt}$ between $T_{mrt.tree}$ and sunlit conditions in the output from SOLWEIG (hereafter referred to as $T_{mrt.solweig}$) for each position for each time step. The comparison take into account shadows from surrounding buildings and existing vegetation (hereafter referred to as $shadows_{solweig}$) and their physical structures, i.e. pixels in $T_{mrt.tree}$ are removed where there are shadows from existing buildings

or vegetation. In short, only sunlit pixels in $T_{mrt.solweig}$ that can be shaded by the generated tree for a given position are used in the comparison. The product from the comparison between $T_{mrt.solweig}$ and $T_{mrt.tree}$ is a new raster with an estimated difference in $T_{mrt}$ between sunlit and tree shade (hereafter referred to as $\Delta T_{mrt}$) for every position in the study area where it is possible to locate a tree. This raster gives a potential decrease in $T_{mrt}$ and is estimated from simply one tree, which means that the estimated potential decrease in $T_{mrt}$ for each position is without any interference of tree shadows from any of the other trees, i.e.

combination of several trees if the model is used to locate optimal positions for more than one tree. The output $\Delta T_{mrt}$ raster from the model initialization is used subsequently in the model to find optimal positions for several trees. To avoid possible boundary effects, five percent of the y extent and x extent are removed at all sides in the input data from SOLWEIG ($T_{mrt.solweig}$ and $shadows_{solweig}$).

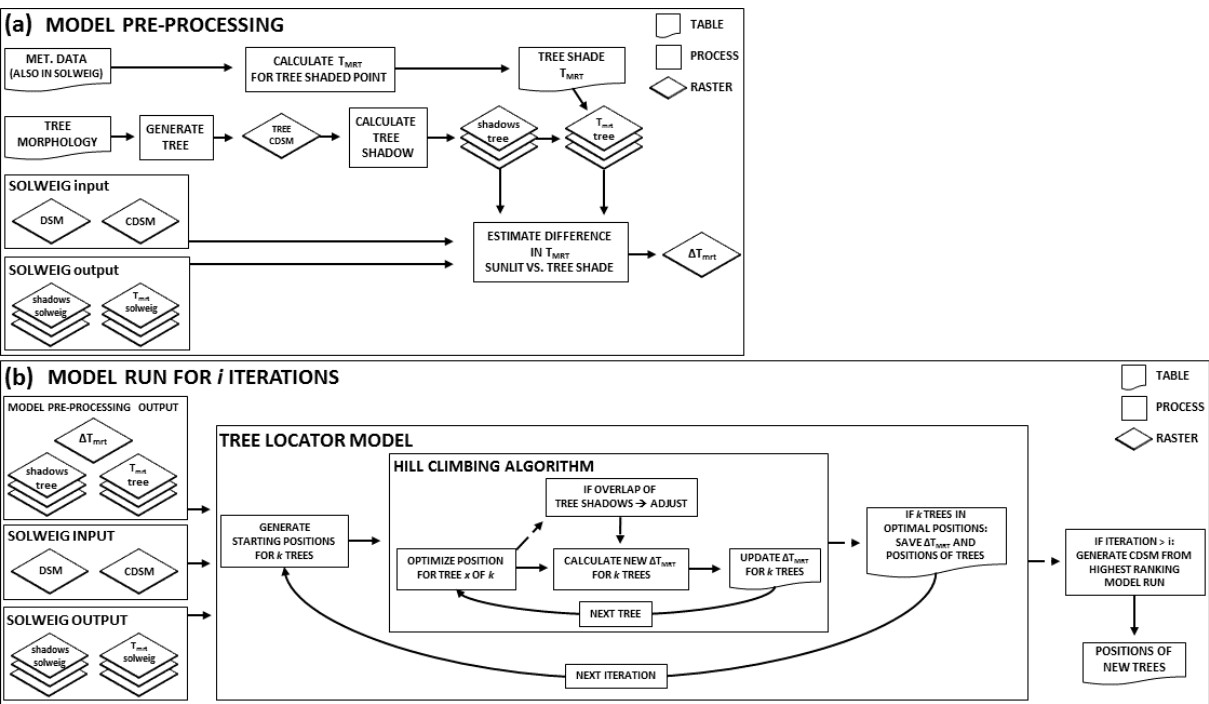

**Figure 1** Flowchart of the model (a) showing the model pre-processing and (b) showing the optimization of tree locations. The flowchart consists of tables (e.g. tree morphology and meteorological data), processes (e.g. GENERATE TREE, which is an algorithm that generates a general 3D tree form from input tree morphology table data) and raster grids. Some grids can have more than one layer, where each layer represents one time step (e.g. TREE SHADOW, where each layer represents shadow pattern for a specific time step). The end product is a table with the positions of the trees.

## 2.2 Algorithms in the model

In the model initialization, described in Sect. 2.1, only one tree is used to calculate $\Delta T_{mrt}$ for every possible location. In theory, it would be possible to calculate $\Delta T_{mrt}$ for any number of trees for all possible combinations of locations. However, brute-force calculations with more than one tree would substantially increase the computational time as the number of combinations would increase exponentially. One way to avoiding brute-force calculations, but still find suitable solutions within limited time or with limited computational power, is the use of metaheuristic algorithms (Luke, 2013). Metaheuristic algorithms are not guaranteed to find the best solution, but nevertheless, they are helpful when brute-force calculations are too extensive and, with a given number of iterations or amount of time, metaheuristic algorithms can lead to a satisfactory result to an optimization problem. The model described in this paper utilizes a hill climbing algorithm (Luke, 2013) to find optimal positions, in combination with one of two ways of assigning starting positions of new trees: random and genetic (Sect. 2.2.1). The basic principle in TreePlanter, with a hill climbing algorithm, is that the $\Delta T_{mrt}$ raster is explored step-wise for better positions, and if shadows$_{tree}$ of two or more trees overlap, adjustments are applied. This means that the $\Delta T_{mrt}$ raster can be explored freely, and the estimated difference in $T_{mrt}$ at each position applies, only to be adjusted if there is an interference, i.e. overlapping tree





shadows. In this way, the number of calculations are decreased extensively. The exploration of the $\Delta T_{mrt}$ raster is conducted until no better positions can be established for the trees, which means that the trees are in their local optimal positions based

on their starting positions. After this, the model restarts a new iteration. The model is set to restart for $i$ number of iterations. The iterations are necessary to initiate new starting positions for the trees. In this way, the trees can avoid finding the same local optimums. Here, a local optimum is defined as a position for a tree, where if it were in any of its neighboring positions, it would have a lower mitigating effect on $T_{mrt}$ (described in more detail in Sect. 2.2.2). Local optimums are, however, not necessarily best positions in the $\Delta T_{mrt}$ raster, as there can be many local optimums within this raster. A large number of

iterations will increase the number of unique starting positions and combinations of starting positions by several trees. This in turn means that more of the $\Delta T_{mrt}$ raster and its local optimums will be explored.

Following is an example of how the hill climbing algorithm can decrease number of calculations, compared to brute-force calculations. An area with $n = 500$ possible locations for trees, where optimal positions for $k = 5$ trees are studied, would require $\frac{n!}{(n-k)!k!} = 2.5 * 10^{11}$ brute-force calculations, considering all possible combinations. Using the hill climbing

algorithm, it is possible to run the model for a given number of iterations. Running it with e.g. $i = 5000$ iterations, where in every iteration each tree would move imaginarily $m = 100$ times, would estimate to approximately $k * m * i = 2.5 * 10^6$ calculations, which is substantially faster in comparison.

A flowchart of the second part of the model, the tree locator, is shown in Fig. 1b. Input data to the model are the $\Delta T_{mrt}$ ($\Delta T_{mrt}$ in Fig. 1b), $T_{mrt.tree}$ and shadows$_{tree}$ rasters from the model initialization, as well as $T_{mrt.solweig}$ and shadow$_{solweig}$ rasters from

SOLWEIG. Pseudocode for the algorithms in Sect. 2.2 is presented in Fig. 2.

```
1. trees = number of trees
2. iterations = number of iterations the model will be run
3. tmrt = decrease in Tmrt

4. for i in iterations:
5.     tree_positions = give trees starting positions (section 2.2.1)
6.     no_better_position = 0
7.     while no_better_position < 1:
8.         for tree in trees:
9.             explore neighbouring positions of tree (section 2.2.2)

10.            check for possible overlap of tree shadows and make necessary adjustments (section 2.2.2)

11.            estimate decrease in Tmrt for surrounding positions

12.            if decrease in Tmrt is higher when tree is positioned in any of the neighbouring positions:
13.                tree_positions = store new position for tree
14.                tmrt = store new higher decrease in Tmrt for trees

16.            if none of trees can find a better position:
17.                no_better_position = 1

18. return tree_positions, tmrt
```

**Figure 2** Pseudocode of the tree locator algorithm in the model.





### 2.2.1 An iteration of the Hill climbing algorithm

The fundamental procedure of the hill climbing algorithm is that the model begins with $k$ number of starting positions for $k$
number of trees, and then cycles through these trees repeatedly to attempt to move trees to a better position. For each tree, the
algorithm searches the adjacent eight pixels for a higher difference in $T_{mrt}$ compared to the tree's current position, and
potentially moves the tree one pixel, and then performs the search for the next tree. When the search/move has been performed
for all trees, the algorithm cycles over all trees again. In this way, the raster is explored until local optimums are determined
for each tree in relation to the other moving trees. If two trees are in such a proximity to each other that their shadows overlap,
the overlap is adjusted for, i.e. the decrease in $T_{mrt}$ is counted only once. This means that shadows can overlap if this would
provide a more favorable shading effect. When all trees have found their optimal positions for an iteration of the hill climbing
algorithm it saves the positions and the corresponding decrease in $T_{mrt}$. A new iteration of the hill climbing algorithm then
commences with new starting positions (Sect. 2.2.2), and the model continues for $i$ number of iterations. In the end of the
model run the iteration with greatest decrease in $T_{mrt}$ is determined and the corresponding tree positions will be used as output.
If two or more tree shadows overlap, an adjustment of the decrease in $T_{mrt}$ is necessary, or the decrease in the overlapping
shadows would be accounted for more than once. Testing for potential overlaps is conducted in different ways in TreePlanter.
A first test is executed by comparing distances between the trees, where large distances can rule out any possible overlap. If
distances, on the other hand, are close enough for potential overlap, additional calculations are executed to determine the exact
number of overlapping pixels, from which adjustments of reduction in $T_{mrt}$ are estimated.
Another functionality in the model, connected to adjustment, is nudging. Nudging is initiated if two or more shadows are
overlapping or next to each other, creating a large continuous shadow. When initiated, it will try to move the trees
simultaneously in the same direction, to see if there are better positions in their vicinity. For example, if two trees have a
combined continuous shadow, it will search their adjacent west pixels simultaneously, then search the northwest pixels,
etcetera, until all eight adjacent directions have been explored. This is to prevent trees from occupying a position, when there
are potentially better positions for them if one of them is relocated.

### 2.2.2 Starting positions for iterations

Two methods to derive starting positions are available in the model: random and genetic. Depending on the size of the model
domain, number of starting positions can be extensive. For example, a 100 x 100 domain with no buildings would have 10000
possible locations, and thus same amount of possible starting positions for the trees. Even if 3000 of these pixels are occupied
by buildings or other hinders, e.g. roads or fountains, and including a buffer zone of one radian of the canopy diameter, there
are still 7000 possible starting positions. The buildings would largely influence the $T_{mrt}$ in some pixels, e.g. on the northern
side of buildings, which would be shaded. Pixels where the decrease in $T_{mrt}$ is zero, i.e. pixels that are already shaded, are
excluded. Furthermore, trees cannot be within one tree canopy diameter of each other when they start, as this would mean that
their canopies would be overlapping.



In the random algorithm for starting positions, each tree is assigned a random starting position in the $\Delta T_{mrt}$ raster at the beginning of each iteration. That is, if the model is executed with 3000 iterations, the trees will start with new random positions in each individual iteration. The algorithm for random starting positions was evaluated against a genetic algorithm for starting positions. Compared to the random starting algorithm, the genetic algorithm inherits starting positions from local optimums of the previous population. This means that starting positions of the next model iteration will be based on the best positions of
the previous iteration. In the first iteration, each tree is assigned a random starting position based on the $\Delta T_{mrt}$ raster. In the second iteration, the positions will be determined randomly from the y and x coordinates of the local optimal positions (local optimums) in the previous iterations. Here, a tree can have its y coordinate from one tree and x coordinate from another tree. Mutation of either the y or the x position will occur if a tree have reached positions without improvement in decrease in $T_{mrt}$ for three consecutive iterations. In this sense, mutation means that the y or the x position (randomly decided which one will
that will mutate) is set to a random position in the planting area. Mutation can also occur if the starting positions for the trees are too close to each other, i.e. the trees have converged to a very confined area and starting positions for a tree are within one diameter of the other trees 50 times in a row.

## 2.3 Summary of the model

This section gives a short summary of the model in a chronological order:
1. SOLWEIG is executed for a given number of time steps based on meteorological data (incoming shortwave irradiance, air temperature and relative humidity) and other necessary gridded input data to simulate $T_{mrt}$ ($T_{mrt.solweig}$) and shadow (shadows$_{solweig}$).

2. A planting area is determined within the spatial extent of the output data from SOLWEIG, e.g. a square in an urban area.
3. A tree form is generated based on tree height, canopy diameter and trunk zone height (height between ground and canopy).

4. The effect of the tree form in step 2 on radiant load is calculated ($T_{mrt.tree}$ and shadows$_{tree}$) based on the same meteorological data as in step 1. This is determined for a flat unobstructed area.

5. Based on the $T_{mrt}$ and shadow patterns from step 1 ($T_{mrt.solweig}$ and shadows$_{solweig}$) and step 3 ($T_{mrt.tree}$ and shadows$_{tree}$)
a difference in $T_{mrt}$ is estimated ($\Delta T_{mrt}$) for each position in the planting area

6. Decide number of trees ($k$) to optimize locations for in respect of $T_{mrt}$ mitigation.

7. Give each tree a random starting position in the gridded planting area.

8. Execute hill climbing algorithm. Adjust for potential overlap by any of the $k$ trees.

9. Save optimal positions of trees and corresponding decrease in $T_{mrt}$.
10. Repeat from step 7 for $i$ number if iterations, with either the random or genetic starting algorithm.

11. Output of tree positions for iteration with greatest decrease in $T_{mrt}$.



### 2.4 Greedy algorithm

As explained in Sect. 2.2, best solution is not known unless brute-force calculations are applied. Therefore, the hill climbing algorithm has been evaluated against a "greedy" algorithm. Zhao et al. (2017) used a greedy algorithm for strategic tree

placement for optimized tree shade coverage to decrease solar exposure on facades and in turn lower building energy use. Although the focus in this paper is on mitigating $T_{mrt}$ and not on lowering building energy use, optimized use of tree shade coverage is the main objective. The results by Zhao et al. (2017) showed that the greedy algorithm determined locations where tree shade coverage was optimized, while simultaneously had low negative effect of tree shade on rooftop solar panels. The greedy algorithm is elemental in that optimal positions for trees are determined one at a time, based on the $\Delta T_{mrt}$

raster described in Sect. 2.1. When a position is determined for a tree, this position and the pixels covering the canopy and a surrounding buffer of one radian of the tree canopy are occupied. Furthermore, spatial patterns of $T_{mrt.solweig}$ and shadows$_{solweig}$ are updated to include spatial $T_{mrt}$ and shadow patterns of the newly added tree, which means that a tree shade of a subsequent tree will not have a mitigating effect in those areas. The greedy algorithm can be described as ranking positions, where the first tree will be located in the position where the highest potential decrease in $T_{mrt}$ can be achieved, the

second tree in the second ranking position (taking into account the shading from the first tree), et cetera. Tree locations determined by a greedy algorithm for mitigation of $T_{mrt}$ are used here to evaluate tree locations determined by the hill climbing algorithm, as the greedy algorithm is expected to always find tree locations with high potential decrease in $T_{mrt}$.

A short summary of the greedy algorithm is as follows:

1.  Step 1-5 in Sect. 2.3.
2.  Determine best position in $\Delta T_{mrt}$ for one tree.
3.  Update $T_{mrt.solweig}$ and shadows$_{solweig}$ to include $T_{mrt.tree}$ and shadows$_{tree}$ based on the position in step 2. Remove all positions within one canopy diameter of the tree location as a future possible location for a tree.
4.  Recalculate $\Delta T_{mrt}$ with the updated $T_{mrt.solweig}$ and shadows$_{solweig}$.
5.  Repeat from step 2 for $k$ trees.

## 3 Model evaluation

### 3.1 Model domain and forcing data

The case study area is taken from the central parts of Gothenburg, Sweden, centered to the square Järntorget (Fig. 3), a hub for public transport. Being a hub for public transport with surrounding blocks occupied by restaurants and small shops, makes it one of the busiest areas in Gothenburg. The southern part of the square was selected for the model evaluation. The square is

intersected by tram tracks seen in the northern and western part of Fig. 3a, with bike lanes in the eastern part. Within the potential planting area, there is a fountain, which is excluded as a possible location for new trees. The input geodata consists



of a DSM, a DEM (Digital Elevation Model) including ground heights and a CDSM (Canopy Digital Surface Model) including vegetation height. Only vegetation higher than 2 meters is present in the CDSM (Fig. 3b). All gridded geodata have a pixel resolution of 1 meter. The geodata originates from the Building and Planning Office in Gothenburg. Hourly meteorological

data (shortwave radiation, air temperature and relative humidity) used were from the nearby Swedish Meteorological and Hydrological Institutes weather station number 92513 (WMO 2513). The meteorological data represents a typical clear summer day in Sweden close to summer solstice (June 22 1983).

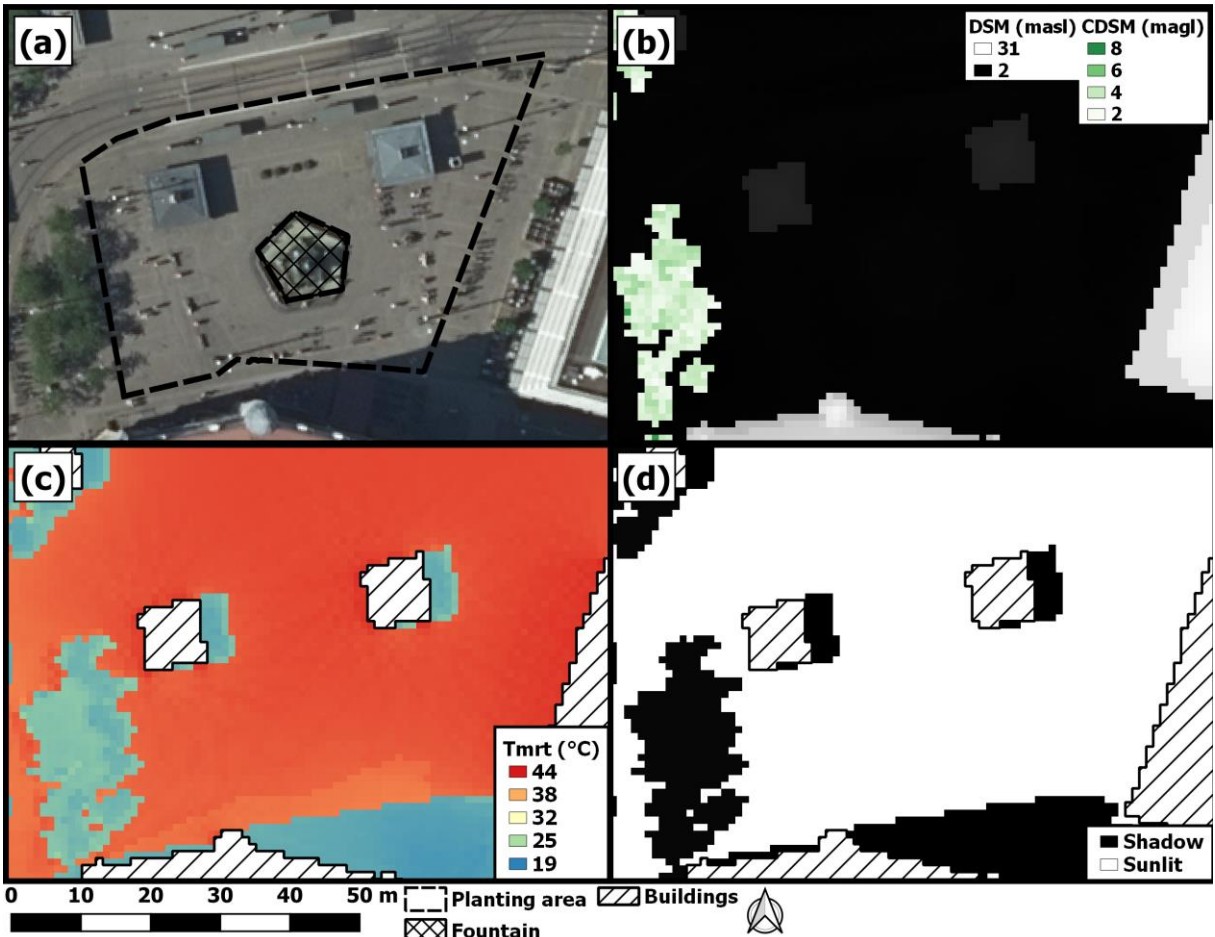

**Figure 3** Map of case study area with (a) Orthophoto RGB 0.25 m © Lantmäteriet where the dashed black line determines the planting area

for possible tree locations and (b) DSM and CDSM. Plots (c) and (d) show examples of output from SOLWEIG, where (c) is $T_{mrt}$ and (d) is shadow patterns for 1700 LST on June 22 1983.

### 3.2 Tree locations generated by TreePlanter

The model was evaluated for 0900-1600 LST and 1300-1600 LST for three different tree sizes (Table 1), as well as an evaluation with four, five and six trees with tree size large.





**Table 1** Table with different tree sizes used for model evaluation.

| Tree size | Tree height (m) | Canopy diameter (m) | Trunk zone height (m) | Transmissivity ($\tau$) |
|-----------|-----------------|---------------------|-----------------------|-------------------------|
| Small | 5 | 3 | 2 | 0.03 |
| Medium | 8 | 5 | 2 | 0.03 |
| Large | 12 | 7 | 3 | 0.03 |

The positions of the trees were determined using the genetic algorithm and 20000 iterations. The raster's with mean $T_{mrt}$ were produced by running SOLWEIG but with an updated CDSM containing the optimized located trees. The difference maps were produced by comparing SOLWEIG outputs of $T_{mrt}$ with and without the optimized trees, i.e. before (sunlit) and after (shaded).

### 3.2.1 Optimal locations for trees with different sizes 0900-1600 LST

Figure 4a shows the locations of five small trees over the period 0900-1600 LST. TreePlanter locates the trees close to the west building. Since the south facing façade of the west building is sunlit most of the time steps during the studied time period, this is where the model identifies optimal positions for the trees. The relatively small trees also cast relatively small shadows, which allow the trees to be located close to each other, as well as close to the building. This results in approximately evenly spaced trees aligned along the south and east facing façades, where their shade allow for a decrease in $T_{mrt}$ of up to 23 °C (Fig.

4b).

Locations for medium trees (Table 1) are shown in Fig. 4c. Here, the trees are more scattered, and optimal positions are established between the two buildings. The area in front of the south facing façade of the west building is now shaded by only one tree, compared to the previous example, where this area was shaded by 2-3 trees. Continuing, the area in front of the east facing façade is now less shaded than in the previous example. On the other hand, there is now an increased shaded area

extending to the east building, because of the greater tree size with their corresponding increased tree shades (Fig. 4d).

When tree size is increased further (large, Table 1), the trees are dispersed even more (Fig. 4e). The area in front of the south facing façade of the west building is now less shaded than in previous examples, and the area in front of the east facing façade is barely shaded at all. Furthermore, two trees are at the northern border, shading areas outside the planting area. Another two trees are shading parts of the fountain. This results in a mitigating effect outside the planting area (Fig. 4f).

**Figure 4** Mean $T_{mrt}$ for 0900-1600 LST on June 22 1983, with locations of (a) five small trees, (c) five medium trees and (e) five large trees in green (see Table 1). (b), (d) and (f) are corresponding differences in $T_{mrt}$ between tree shade and sunlit for (a), (c) and (e) respectively. The positions are determined with the genetic starting algorithm and 20000 iterations.





### 3.2.2 Optimal locations for trees with different sizes 1300-1600 LST

In the previous example (Fig. 4) all individual time steps for 0900-1600 LST and their respective spatial shadow and $T_{mrt}$ patterns are integrated to study and analyze tree locations over a longer time period. However, excessive radiant load and heat stress is in general most pronounced during early afternoon, when solar radiation potentially is high, and surrounding heated surfaces emits large amounts of longwave radiation. Therefore, the model was run for 1300-1600 LST, with same tree sizes as in Fig. 4, to study the tree locations during this time period. The corresponding results are presented in Fig. 5.

The results from the model run with small trees (Table 1) are shown in Fig. 5a. One striking difference compared to Fig. 4a is that all trees except one are concentrated around the east building instead of the west building. One tree end up shading the area in front of the south facing façade of the west building. In Fig. 5b it is possible to see a decrease in mean $T_{mrt}$ of up to 26 °C for almost all areas shaded by the trees.

When tree size is increased to medium all trees end up in the eastern part of the study area (Fig. 5c). The locations established

by the model provide shade around the entire previously sunlit area of the east building, and a large decrease in $T_{mrt}$, as seen in Fig. 5d.

In the last example, with the tree size large (Fig. 5e), the trees are more or less in the same locations as in Fig. 5c, and similar assumptions can be made as in the previous example. However, they are now positioned slightly further south, as their shadows extend further because of the higher tree height. It is also visible that they are farther apart, as the diameter of the tree canopy

is larger, and thus the tree shade increases in width as well. In this example, however, the area shaded by the trees is now continuous for all trees (Fig. 5f), compared to Fig. 5d. That is, the area where the tree shadows have an effect covers almost all areas along the south and western sides of the east building.





**Figure 5** Mean T$_{mrt}$ for 1300-1600 LST on June 22 1983, with locations of (a) five small trees, (c) five medium trees and (e) five large trees in green (Table 1). (b), (d) and (f) are corresponding differences in T$_{mrt}$ between tree shade and sunlit for (a), (c) and (e) respectively. The positions are determined with the genetic starting algorithm and 20000 iterations.



### 3.2.3 Optimal locations depending on number of trees

One aim with TreePlanter and the hill climbing algorithm is to see if the combined and continuous shadow of several trees could influence the positioning of trees. In other words, can e.g. the combined and continuous shadow of two trees shade an area equivalent to that shaded by one tree, but from different positions. To investigate this, the model was executed with varying number of trees (four, five and six) with tree size large (Table 1). The results from the model run with four trees (Fig. 6a), show that the tree shading the area in front of the southwest facing corner in Fig. 5e is now missing. Furthermore, the westernmost tree is located slightly more south. Other than that, no large difference is visible. The missing tree, of course, has an effect on the amount of area with a decrease in $T_{mrt}$, as seen in Fig. 6b. When increasing the number of trees to six, it is possible to see that the three trees shading the south and west facing façades (Fig. 6c) are in same positions as in Fig. 5e. However, when comparing with figure 6a, only one tree is in the same position (the central one in fig. 5a, shading the south facing façade). All other trees are in different positions. The trees in the western part, along the border of the planting area, are now located further south and only the very north part of this border is still sunlit. For all model runs it is possible to see a large decrease in $T_{mrt}$ (Fig. 5f, 6b and 6d).

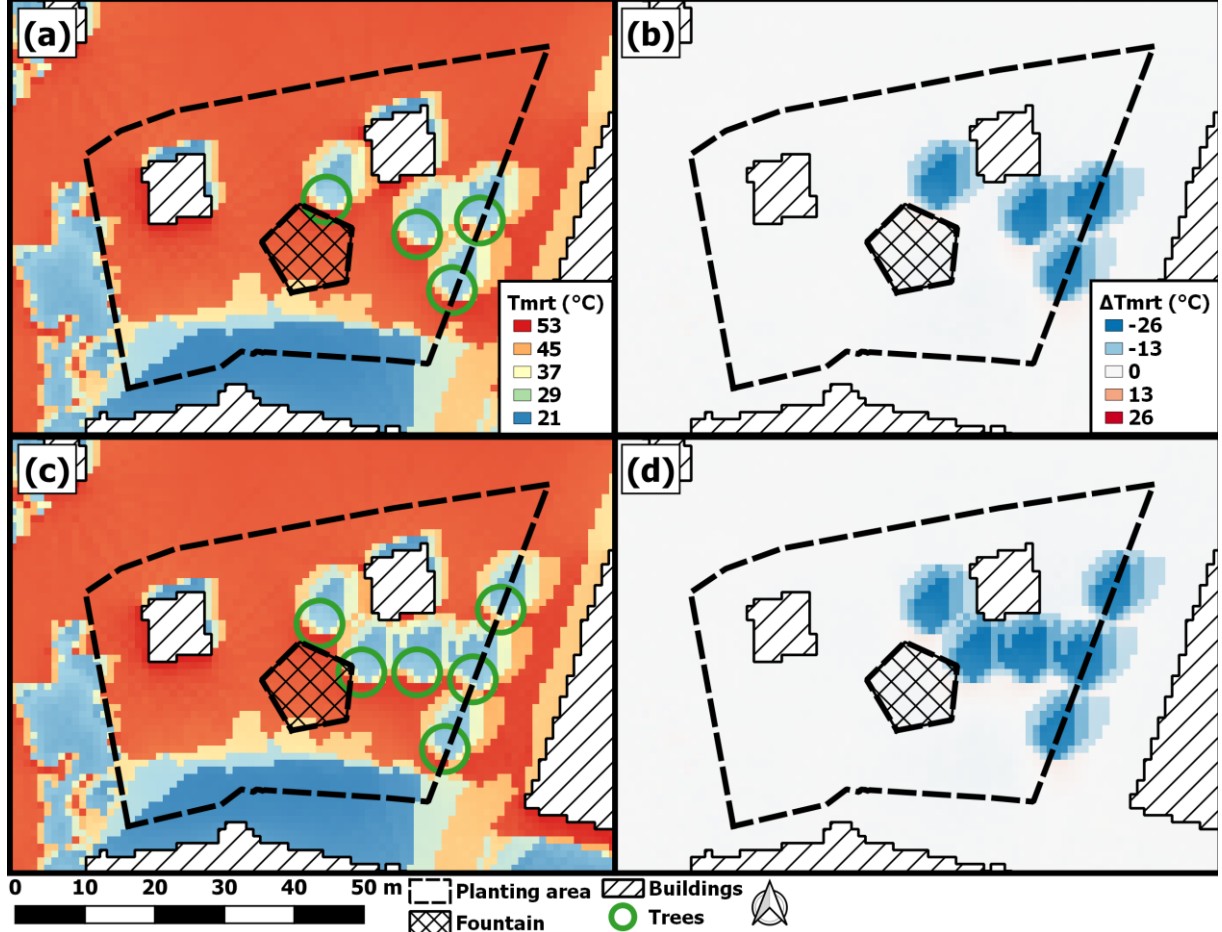





**Figure 6** Mean $T_{mrt}$ for 1300-1600 LST on June 22 1983, with locations of (a) four large trees and (c) six large trees in green (see Table 1 for detailed tree sizes). (b) and (d) are corresponding differences in $T_{mrt}$ between tree shade and sunlit for (a) and (c). The positions are determined with the genetic starting algorithm and 20000 iterations.

## 4 Model performance

The model performance and runtime are dependent on a combination of spatial extent and the pixel resolution of the study area, i.e., the number of model grid points within a domain. Factors such as tree size, number of trees and time steps also affect model performance. The model was executed with the same model domain and forcing data as in Sect. 3.1, but with varying tree size, number of trees, time steps, model iterations and domain pixel resolution, to investigate these dependencies. In the performance analysis a ratio of the potential decrease in $T_{mrt}$ between the hill climbing algorithm and the greedy algorithm was

used to quantify the mitigation benefits of the hill climbing algorithm. The model performance tests were executed on an Intel Core i7-7700 CPU @ 3.60 Ghz with 16 GB RAM @ 2400 Mhz. Figure captions include number of potential locations for trees, which differs depending on tree size due to the fact that a tall tree generates a more extensive ground shadow compared to a short tree. Mean model runtime (s) for all model performance tests are presented in Table 2. Initialization time is excluded and disk I/O is negligible.

**Table 2** Table showing mean model runtime (s) for 100 iterations for two starting algorithms (random and genetic) with different time periods, number of trees and tree sizes. The difference in model runtime (%) corresponds to a change in model runtime with the genetic starting algorithm compared to the random algorithm. A negative (positive) value corresponds to a decrease (increase) in model runtime with the genetic starting algorithm.

| Time period | Trees | Tree size | Random | Genetic | Difference (%) |
|---|---|---|---|---|---|
| 0900-1000 | 5 | Small | 8.4 | 4.9 | -42.1 % |
| 0900-1000 | 5 | Large | 10.0 | 7.7 | -23.3 % |
| 1300-1600 | 5 | Small | 9.7 | 6.1 | -36.9 % |
| 1300-1600 | 5 | Medium | 11.6 | 8.3 | -28.7 % |
| 1300-1600 | 5 | Large | 11.8 | 9.6 | -19.1 % |
| 0900-1600 | 5 | Small | 14.1 | 12.7 | -9.7 % |
| 0900-1600 | 5 | Medium | 26.5 | 19.3 | -27.0 % |
| 0900-1600 | 5 | Large | 38.8 | 25.9 | -33.3 % |
| 1300-1600 | 2 | Large | 1.7 | 1.0 | -40.5 % |
| 1300-1600 | 3 | Large | 4.1 | 2.7 | -34.1 % |
| 1300-1600 | 4 | Large | 7.6 | 5.3 | -30.2 % |
| 1300-1600 | 5 | Large | 11.8 | 9.6 | -19.1 % |
| 1300-1600 | 6 | Large | 16.7 | 17.7 | 6.5 % |
| 1300-1600 | 7 | Large | 22.0 | 36.0 | 63.9 % |
| 1300-1600 | 8 | Large | 29.0 | 72.7 | 150.6 % |





### 4.1 Tree size

All three tree sizes in Table 1 were used for the performance analysis on tree size. The model was run with five trees for each tree size, and both starting algorithms (random and genetic) and 10, 100, 500, 1000, 2000, 3000, 10000 and 20000 iterations. The results from the performance analysis on tree size for seven hourly time steps (0900-1600) LST is shown in Fig. 7. The results indicate that the hill climbing algorithm have established positions with high mitigating potential already after five iterations. After 100 iterations, all trees are at positions similar to or better than the greedy algorithm, regardless of starting

algorithm. However, some divergence can be seen, for example for tree size small and 3000 iterations.

Noticeable in Table 2 is that a larger tree size decreases the speed of the model, but with 5 trees the genetic starting algorithm is faster compared to the random starting algorithm. The difference in model runtime between the two starting algorithms also increases with tree size, from -9.7 % with small trees to -33.3 % with large trees.

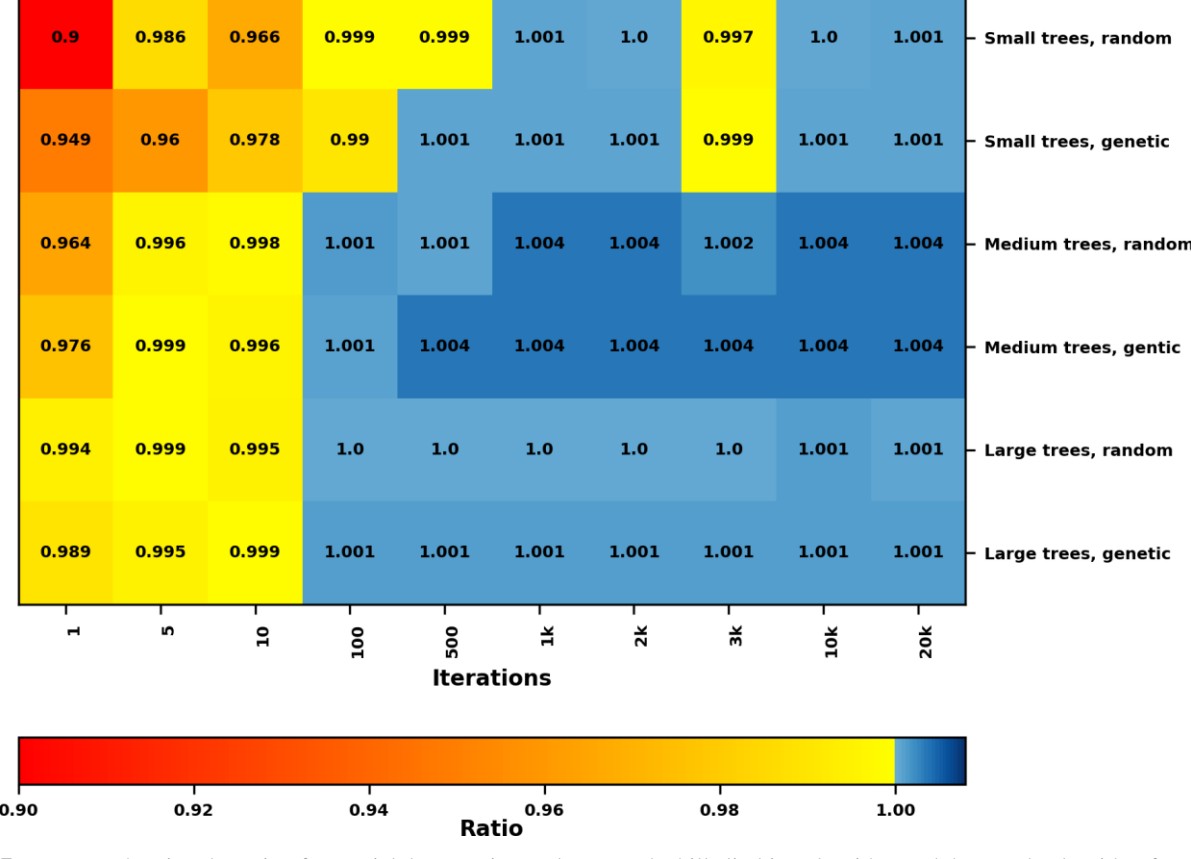

**Figure 7** Heat map showing the ratio of potential decrease in $T_{mrt}$ between the hill climbing algorithm and the greedy algorithm for the three different tree sizes (small, medium and large; see Table 1) for different number of model iterations, for 0900-1600 LST on June 22 1983. Two starting algorithms were used; random and genetic. Each model run was executed with five trees. Potential locations for trees are 1709 for small trees, 1603 for medium trees and 1481 for large trees. A ratio > 1 indicates a larger $T_{mrt}$ decrease with the hill climbing algorithm. Note that cell color where ratios ~1.0 is determined using an extended number of decimal places.



## 4.2 Number of trees

An increase in number of trees increases the complexity of the model and influences the performance and speed as the probability of overlapping tree shadows would increase, regardless of tree size. The model was executed with two, three, four, five, six, seven and eight trees with tree size large (Table 1), with three time steps (1300-1600 LST), with corresponding results presented in Fig. 8. As illustrated, the potential decrease in $T_{mrt}$ is high after 5-10 iterations, and after 500 iterations mitigation benefits for position with the genetic starting algorithm always exceed those from the greedy algorithm. There is also a tendency for a higher ratio with a higher number of trees.

Model speed decreases with an increase in number of trees for both random and genetic, as shown in Table 2. The speed performance of the genetic algorithm outcompetes the random algorithm for all runs except six, seven and eight trees. With six, seven and eight trees model runtime increases, for eight trees quite extensively with a difference of around 150 % between the random and genetic with 20000 iterations.

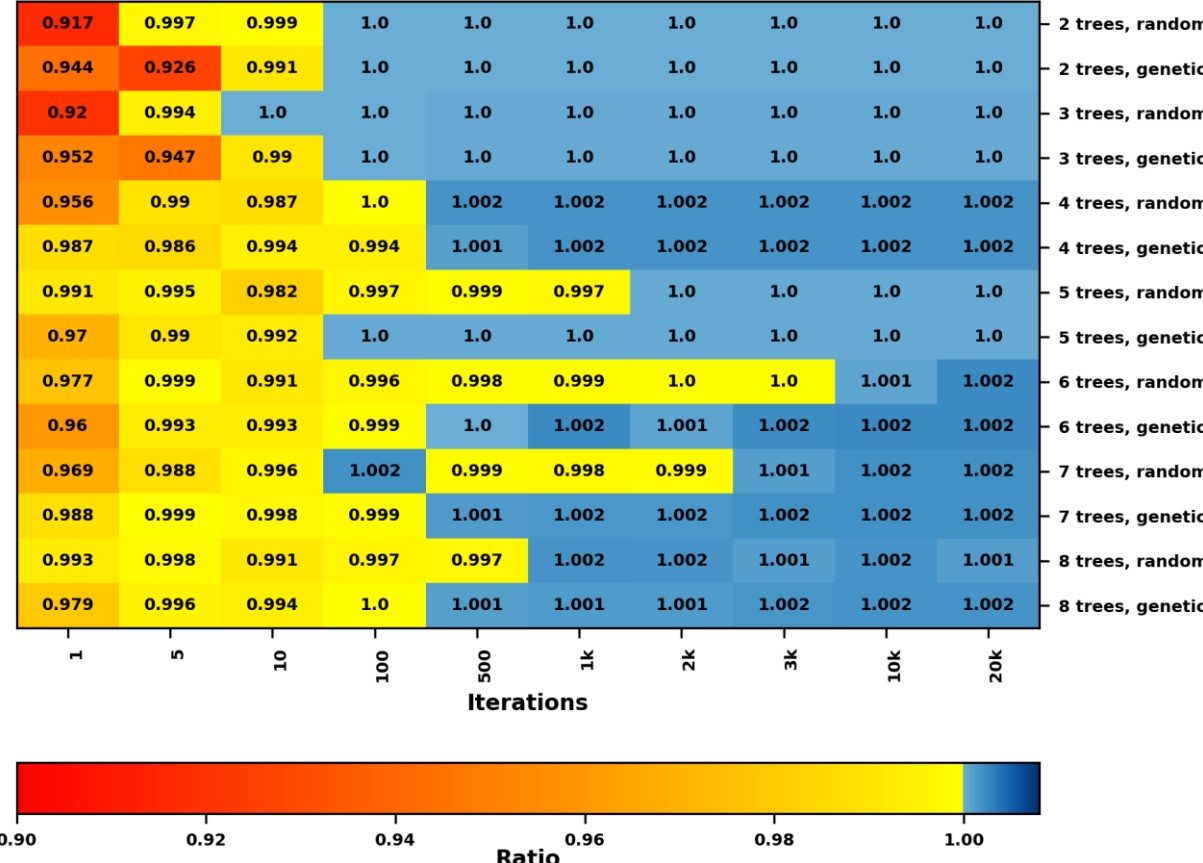

**Figure 8** Heat map showing the ratio of potential decrease in $T_{mrt}$ between the hill climbing algorithm and the greedy algorithm for two, three, four, five, six, seven and eight large (Table 1) trees for different number of model iterations, for 1300-1600 LST on June 22 1983. Two starting algorithms were used; random and genetic. Potential locations for trees are 1481. A ratio > 1 indicates a larger potential decrease with the hill climbing algorithm.





### 4.3 Time steps

The model was also tested for different time steps; one (0900-1000 LST), three (1300-1600 LST) and seven (0900-1600 LST),
to analyze performance and speed, with two different tree sizes: small and large (Table 1).

The time-step performance analysis for small trees (Fig. 9a), similar to previous examples, found positions with high potential
380  decrease in $T_{mrt}$ relatively fast, after 100-500 iterations, and with the genetic starting algorithm, positions are always better
compared to the greedy algorithm after 500 iterations (ratio > 1.0), with the exception of 3000 iterations 0900-1600 LST. For
the random starting algorithm, however, some anomalies were found for 0900-1000 LST and 0900-1600 LST and 2000 and
3000 iterations.

Considering speed with tree size small, there is a large increase in model runtime for the longer time period (0900-1600 LST)
compared to the two other model runs as seen in Table 2 (0900-1000 and 1300-1600 LST). Furthermore, differences in speed
between genetic and random are highest with one time step (-42.2 %)

The time-step performance analysis for large trees (Fig. 9b) found adequate positions after 100 iterations, but similarly to the
small trees, there are some anomalies with the random starting positions, e.g. for 100 iterations 0900-1000 LST and 100, 500
and 1000 iterations 1300-1600 LST.

When analyzing model runtime, it is clear that this increases with number of time steps, similar to that of small trees. However,
largest difference between starting algorithms is with seven time steps (0900-1600 LST, -33.3 %).

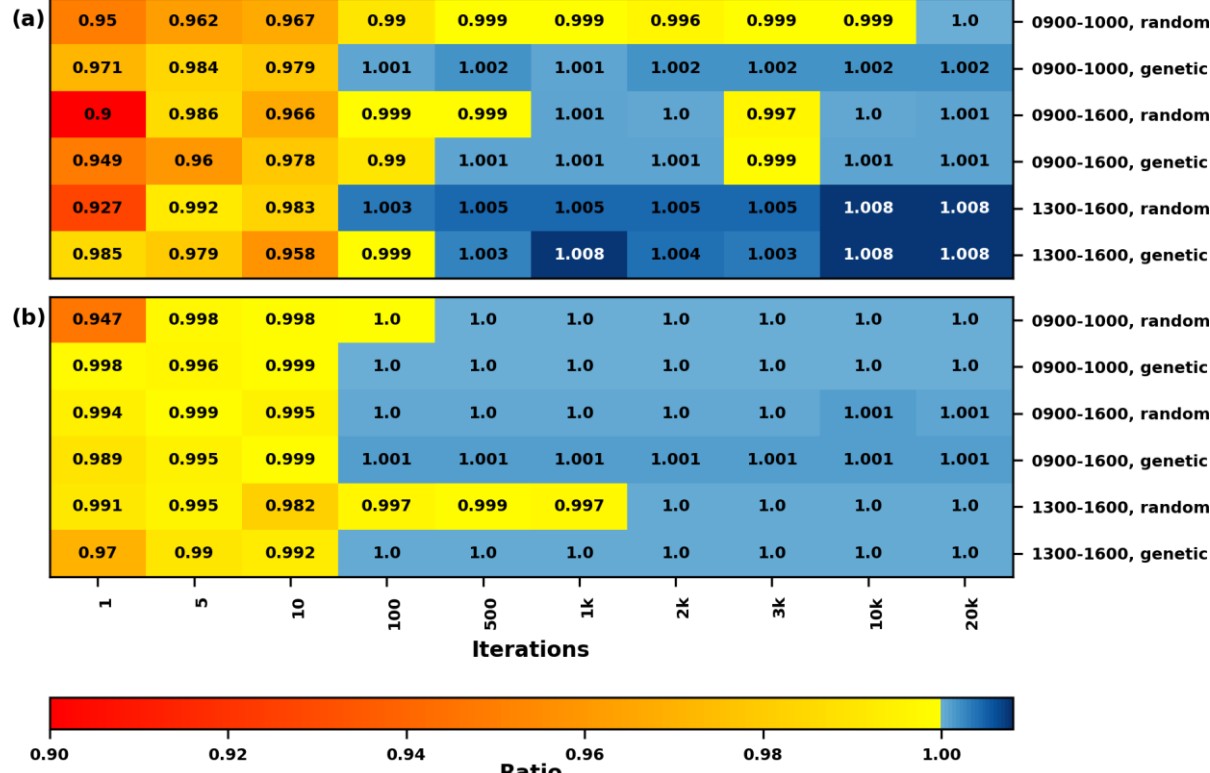





**Figure 9** Heat maps showing the ratio of potential decrease in $T_{mrt}$ between the hill climbing algorithm and the greedy algorithm for (a) five small trees with different time steps and (b) five large trees with different time steps, on June 22 1983 (see Table 1 for detailed tree sizes).
Two starting algorithms were used: random and genetic. Potential locations for trees are 1709 for small trees and 1481 for large trees. A ratio > 1 indicates a larger potential decrease with the hill climbing algorithm.

### 4.4 Model domain size

A last model performance analysis was conducted with special attention to model domain size, here represented by changing pixel resolution. Four different pixel resolutions were evaluated: 2, 1, 0.5 and 0.25 meters. The pixel resolution was tested for
three trees with tree height = 10 meters, canopy diameter = 5 meters and trunk zone height = 3 meters. As expected with a 2D modelling approach, the model runtime increased exponentially with higher pixel resolution, from 17 seconds with a 2 meter pixel resolution to 44, 185 and 1290 seconds for 1, 0.5 and 0.25 meters, respectively.

### 4.5 Tree locations – Hill climbing algorithm vs. Greedy algorithm

As shown in the results for the performance analysis, the hill climbing algorithm, with a high number of iterations, gives equal
or marginally higher potential decrease in $T_{mrt}$ compared to the greedy algorithm. This suggests that the resulting locations for trees are different in the two algorithms. This is illustrated in Fig. 10. The examples from the hill climbing algorithm are with 20000 iterations, for which the hill climbing algorithm always determined positions that had higher potential decrease in $T_{mrt}$ than the greedy algorithm.

In all cases, some trees are in the same locations for both the hill climbing and greedy algorithms, but the hill climbing
algorithm clusters the trees more closely together than the greedy algorithm. In Fig. 10a, the locations for the two western trees are the same, two central trees are only slightly different, but the greedy algorithm places a tree near the east building rather than near the western building. Similar observations can be made in Fig. 10b-d, with some locations the same, some similar, and one tree placed differently. In Fig. 10d the most interesting difference is in the east part of the planting area. Here, the greedy algorithm finds an optimal position in between the two trees found by the hill climbing algorithm, and places a tree by
the west building instead. Comparable results are visible in Fig. 10e, where one of the tree locations by the greedy algorithm is at the very northwest corner of the planting area.



**Figure 10** Locations for trees from the hill climbing algorithm with the genetic starting algorithm and 20000 iterations, and the greedy algorithm. The left column (figures a, c and e) and the right column (figures b, d and f) are for 0900-1600 LST and 1300-1600 LST, respectively, on June 22 1983. Figures (a) and (b) are with small trees, (c) and (d) with medium trees and (e) and (f) with large trees. The underlying map shows mean $T_{mrt}$ for the respective time periods from which the locations were determined.



## 4.6 Occurrences of tree positions

Occurrences maps, showing where trees were located after each iteration in a model run with the random starting algorithm and 20000 iterations, are shown in Fig. 11. The preferred positions are relatively warm, where mitigation from tree shade is high. The general pattern is that preferred positions lie in arcs around one tree-diameter from buildings, along the northern and eastern borders of the planting area, or close to fountain. Within these regions there are a few highly-preferred pixels. Furthermore, south of the buildings, two rows of preferred positions can be seen in almost all figures. The highest occurrence is for small trees in front of the west building (72.6 %).







**Figure 11** Occurrence maps showing percentage of times pixels were found to be an optimal tree position based on model runs with five trees, the random starting algorithm, and 20000 iterations. Grey indicates zero occurrences. Maps (a) and (b) are for small trees, (c) and (d) are for medium trees and (e) and (f) are for large trees. The left column (a, c and e) is for 0900-1600 LST and the right column (b, d and f) is for 1300-1600 LST.



## 5 Discussion

The aim of the TreePlanter is to find locations for the trees where they would have optimal mitigating potential with respect to $T_{mrt}$. The general perception of the results in Sect. 3 implies that this is the case. Most of the trees end up shading areas in front of sunlit walls, which are known as exposed to high radiant load (Thorsson et al., 2011, Lindberg et al., 2016; Wallenberg et al., 2020). During the hottest time of the day in the case study, $T_{mrt}$ under the trees dropped with as much as 26 °C from optimized positioning of trees (1300-1600 LST, Fig. 5). Such a sharp drop can have a profound effect on thermal (dis)comfort,

making an area available for pedestrians, with less negative effects on e.g. health.

However, some differences in tree locations were found, depending on time of day, tree size and number of trees. Analyzing time of day, it possible to see that trees are, in general, located in the western part of the planting area during the longer time period (0900-1600 LST) in Fig. 4, whereas they end up in the eastern part during the shorter time period (1300-1600 LST) in Fig. 5. The differences in locations for trees are explained by spatial $T_{mrt}$ patterns. The hours before noon would increase the

amount of sunlit areas in front of west facing facades integrated into TreePlanter. Thus, mitigation in these areas becomes more important. This shows the importance of timescale for planning of tree locations, and in this sense, season could also be an interesting aspect. Konarska et al. (2014), for example, discussed the importance of deciduous trees for mitigation of high radiant load in summer, as deciduous trees would allow higher transmissivity of solar irradiance in winter when leaves have dropped. Nevertheless, deciduous trees would block approximately 50 % of the incoming shortwave irradiance without leaves

(Konarska et al., 2014). This means that optimized locations for trees in summer could have negative effect on thermal comfort in winter.

Tree size was also found to influence tree locations. The main finding is that, when tree size and hence shadow size were increased, the tree locations were dispersed as the model strives for the largest mitigating effect, which is mainly achieved by utilizing the entire tree shadows. This is most evident in Fig. 4 (0900-1600 LST), where small trees are aligned along the west

building, but medium and large trees are dispersed, covering areas in between the buildings. This leads to two trees, with tree size large, mitigating radiant load outside the planting area, where there in reality are tram tracks (Fig. 4e-f). The effects of tree size on tree locations is also visible in Fig. 5 (1300-1600 LST). In this case, however, it is rather that the shadow has a higher effect in a different location depending on tree size. One of the small trees was positioned in front of the west building. When tree size was increased to medium or large, tree shadows of all five trees have enhanced effect at the east building. This

suggests that a larger tree shade, as the ones in Fig. 5c, is more beneficial in different locations compared to smaller tree shades, and smaller exposed areas can remain exposed. This is important because if mitigation of high radiant load is desired in an area such as the one presented here tree size is decisive. Tree size in this sense can also resemble stages in the tree's life, i.e. juvenile or mature. Thus, it can be important for planners and others to keep in mind that an optimal position for a tree might differ depending on the age (size) of the tree.

Tree locations also changed depending on the number of trees. This is in clear contrast to the greedy algorithm, where tree positions are fixed when a new tree is added. TreePlanter enables tree locations to be influenced by each other, which means





that none of the locations are fixed while the hill climbing algorithm is still cycling through the trees, even if a single tree has reached an optimal position. This is realized with the nudging effect, which will enable two or more trees to explore if their combined and continuous shadow can find a more favorable and efficient mitigation of $T_{mrt}$. An example of this is apparent in
Fig. 5e-f and Fig. 6, where the trees at the east border of the planting area have different locations with four (Fig. 6a-b) and five (Fig. 5e-f) trees compared to six trees (Fig. 6c-d). Another example of how nudging and non-static locations are important when optimizing positions for trees is when comparing positions between the hill climbing algorithm and the greedy algorithm, shown in Fig. 10. Here, it is evident that static locations, as with the greedy algorithm, can result in scattered tree locations. Even though locations established with the greedy algorithm are very efficient for mitigation of high $T_{mrt}$, they are inherently
different compared to those established with the hill climbing algorithm. The locations established with the hill climbing algorithm not only result in higher $T_{mrt}$ mitigation (although this difference is small), but more importantly they are closer together. The occurrences map in Fig. 11 show that the locations established with the greedy algorithm are also found by the hill climbing algorithm. However, with the nudging function, allowing a continuous shadow of several trees to explore the $\Delta T_{mrt}$ raster, space is used more efficiently. Concluding, from a planning perspective, it can be noteworthy that these two
algorithms (hill climbing and greedy) can result in considerably different tree locations.

In the model performance analysis, in Sect. 4, potential decrease in $T_{mrt}$ with the hill climbing algorithm and its two starting algorithms were evaluated against potential decrease in $T_{mrt}$ with the greedy algorithm. The greedy algorithm was expected to always find tree locations that provide high (even if not optimal) $T_{mrt}$ mitigation. This expectation is supported by the results presented here, for example in Fig. 10, where tree locations determined by the greedy algorithm are always in areas exposed
to high radiant load. Zhao et al. (2017), similarly, demonstrated optimized tree shade coverage with a greedy algorithm, but to decrease solar exposure on facades. Thus, the ratio between the two (hill climbing and greedy algorithms) give a good estimation of the performance of the hill climbing algorithm in this sense. From the results in Fig. 7-9 it is evident that the hill climbing algorithm finds acceptable locations already after one iteration as none of the results show a ratio lower than 0.9. After 100-500 iterations, tree locations established by the hill climbing algorithm show an equal or higher mitigating effect
than the greedy algorithm, in almost all cases. There are, however, some exceptions, especially with the random starting algorithm. Running the hill climbing algorithm with the genetic algorithm, on the other hand, results in tree locations with a mitigating effect equal to, or higher, than the greedy algorithm after 500 iterations for all cases except 0900-1600 LST with small trees with 3000 iterations. Running the model with 20000 iterations, on the other hand, always resulted in tree locations with equal or higher mitigating effect compared to the greedy algorithm, regardless of starting algorithm. Furthermore, the
exceptions, seen for example in Fig. 7 with small trees and 3000 iterations, is an example of the fact that, as explained in Sect. 2, metaheuristics are not guaranteed to find an absolutely optimal solution to a problem. Adding to this, it is not possible to prove that any of the locations established by the hill climbing algorithm are absolutely optimal, unless extensive and computationally demanding brute-force calculations are performed.

The results from the comparison between the genetic and random starting algorithms are in line with those by Stojakovic et al.
(2020). Although their approach was different from the one presented in this paper in that they did not use a hill climbing





algorithm, they did use a genetic algorithm. In their method, tree locations were determined from tree locations from a previous iteration. Their results showed that tree locations with the genetic algorithm had a higher mitigating effect compared to randomly positioned trees. This can be compared to the results found in this paper, where tree locations with a high mitigating effect are sometimes found with less iterations with the genetic algorithm compared to the random algorithm. Adding to this,

Stojakovic et al. (2020) found convergence after approximately 3000 iterations. Here, some of the examples show convergence already after 500 iterations, indicating that exploration with the hill climbing algorithm in combination with a genetic algorithm for starting positions could be a beneficial approach.

When model runtime was evaluated, it was quite clear that when time steps, tree size or number of trees increased, so did model runtime (Table 2). When time step was increased, the number of necessary calculations increased, mainly when tree

shadows overlap. This was further reinforced when tree size was increased. An increase in tree size increases the area shadowed by the tree, regardless of time of day, as long as the solar elevation is above the horizon. A larger shadow increases the possibility of tree shadows overlapping, and overlapping shadows need adjustment to find optimal locations. Increasing the number of trees also adds to the probability of overlapping shadows. Increasing model domain size (by changing pixel resolution) resulted in an exponential increase in model runtime, from very fast for low resolution (2 meter) to relatively long

model runtime with a high resolution (0.25 meter). As model pixels become smaller, the number of possible locations for trees increases substantially, leading to increased model runtime. The four factors described above all influence the model complexity and computational time. Largest influence comes from the number of time steps. The increase in computational time is considered to be an effect of overlapping tree shadows that need adjustment. Increasing tree size and the number of trees further increases the possibility of overlapping shadows. Additionally, an increase in number of time steps would increase

the time to adjust, as well as a larger shadow of integrated time steps. Moreover, larger model domains can substantially increase model runtime.

When evaluating and comparing the two starting algorithms, the genetic algorithm improved model runtime in all cases except when number of trees was increased to six, seven and eight, for which model runtime increased. On the other hand, the genetic algorithm determined optimal positions at an earlier stage compared to the random algorithm. This, however, needs further

testing, but the enhanced speed of the genetic algorithm can reasonably be explained by two factors. One being that warm areas, i.e., local optimums with high $T_{mrt}$ are in close vicinity to each other, which means that subsequent generations (iterations) potentially start closer to local optimums compared to random starting positions. The other factor could be the geometry of the urban setting studied here, where sunlit areas follow the x-axis or the y-axis. As an example, the two small buildings inside the study are at approximately same y-positions, with the right hand being located at a slightly higher position.

The area in between the buildings is located at approximately same x-position. Thus, if a tree inherits the y-position from a tree that in the prior run ended up in front of one these buildings, it would probably not have to explore too many pixels in order to a find a new local optimum. Likewise, if a tree inherits its x-position from a tree with a local optimum between the two buildings, it would not have to move far to find optimal position. The random starting algorithm, on the other hand, could have trees starting in positions where exploration of the $\Delta T_{mrt}$ raster would take longer in order to find a local optimum. These



factors can possibly also explain the decrease in model runtime with the genetic starting algorithm when number of trees were set to seven or eight. The trees would start closer to each other, and chance of overlapping shadows is increased.

As described above, model runtime differs depending on tree size, number of trees, time steps and size of model domain. With the genetic starting algorithm, for five small trees giving 1709 possible locations seven time steps (0900-1600 LST), each iteration takes approximately 0.13 s. Using same setup, but with the large tree size and 1481 possible locations, each iteration

takes approximately 0.26 s. For comparison, Stojakovic et al. (2020) utilized Rhinosceros 3D (CAD modeling software), Grasshopper (add-on to Rhinosceros 3D), and an evolutionary algorithm (inheritance) from Galapagos (a plugin for Grasshopper with generic solvers (Rutten, 2013)). With their setup, for five trees (two sizes simultaneously, three with a canopy diameter of 8 m and two with 17 m), ten time steps and 625 possible locations for trees, they had a mean iteration runtime of around 9 s. This suggests that the 0.26 s for five large trees and seven time steps in the model presented in this paper

is highly efficient.

Genetic algorithms have been used successfully in previous research on optimization problems e.g. locations for hospitals in Hong Kong (Li and Yeh, 2005), locations for train stations in Leicester, UK (Ahmed et al., 2013), spatial land use allocation planning in Guitiriz, Spain (Porta et al., 2013) and tree locations (Stojakovic et al., 2020). The results presented in this paper show how random starting positions and a genetic algorithm to determine starting positions can be used together with a hill

climbing algorithm for optimized tree locations. The benefit of using a hill climbing algorithm is that it enables thorough exploration of potential tree locations and at the same time simplifies any potential adjustment for potentially overlapping tree shadows. Using the genetic algorithm to determine starting positions, in most cases, improved model runtime and convergence compared to randomly determined starting positions.

Even if model runtime is relatively fast, it needs improvement for the hill climbing algorithm to become a generic tree location

finding tool. Real world applications will likely have more trees and larger areas of interest than this study. This will require an increase in the number of iterations, as the number of possible locations would increase. Running TreePlanter with many iterations, several trees and in a large area is time consuming. The examples shown here are relatively fast. However, as seen with the genetic algorithm and six, seven or eight trees, model runtime increases quite extensively. A greedy algorithm, as an option to the hill climbing algorithm, is included in TreePlanter to support larger studies (see Sect. 5.2).

**5.1 Model limitation and potential**

In SOLWEIG, as well as TreePlanter presented in this paper, there is no option for individual tree parameterization. For example transmissivity can be set to one general value for all trees. The default in SOLWEIG is 3 % for summer (Konarska et al., 2014). Continuing, the model and the examples presented in this paper only include radiant load. However, there are other factors that can be affected from adding more trees, e.g. wind and evapotranspiration, which are not included. While

evapotranspiration from trees has been shown to have a negligible effect on the thermal comfort, trees affecting pedestrian-level wind can have a large effect on outdoor thermal comfort (Lee and Mayer, 2020). There are also other factors, connected



to plant physiology, that need consideration when planting a tree in an urban area such as root spacing, soil conditions, climatic growing conditions and water availability (Vogt et al., 2017), which are not examined here.

Not included in the current version of TreePlanter is a recalculation of sky view factor between iterations. Omitting
recalculation of SVF has a minor effect on the total radiant conditions under the tree as well as the influence on the radiant conditions in the surrounding environment, when the moving trees block parts of the sky. View factors influence e.g. the amount of diffuse shortwave irradiance reaching an area as well as influencing the longwave irradiance under and around the trees. These effects are relatively small, however, whereas including recalculations in the model would increase runtime substantially. We thus do not foresee including SVF recalculations in the model. See Lindberg and Grimmond (2011) for a
more detailed discussion.

Speak et al. (2020) studied the shading effect of single trees in Bolzano, Italy. They found that leaf area density and canopy diameter are key in mitigating surface temperatures, and concluded that strategic planting of urban trees with cautious selection of species can help mitigate surface temperatures. Antoniadis et al. (2020), similarly, discussed strategic planting and positioning of trees, to alleviate heat stress in urban schoolyards. The model presented in this paper can aid in the strategic
planting of trees based on $T_{mrt}$, as it enables the possibility of positioning trees and changing canopy diameters, tree height and trunk zone height. The possibility of changing transmissivity of shortwave irradiance through the canopy can be used as an analogy for leaf area density. Furthermore, it allows for analysis of days with different meteorological data, and even a possibility of combining days from different seasons. Moreover, with modifications, the methodology described in this paper could be used for other optimization purposes, e.g., mitigation of incoming solar irradiance with respect to harmful UV.
Future possible developments include possibility to locate positions using trees of different sizes simultaneously. Furthermore, studies on the effects of locations during different weather conditions and seasons are expected.

## 5.2 Tool accessibility

To facilitate use and accessibility to other researchers and service providers, TreePlanter is available as a tool as part of the UMEP climate service tool in the open-source geographical information system QGIS (https://qgis.org). The Graphical User
Interface (GUI) from the Processing Toolbox in QGIS is presented in Fig. 12. For simplification, the pre-processed input data required from SOLWEIG for TreePlanter will be provided through a tick box in the SOLWEIG GUI.

Moreover, a vector polygon layer that determines a *planting area* where it is possible to plant trees within the extent of the SOLWEIG output is required (see first paragraph in Sect. 2). This option can be used to avoid roads, statues, water bodies or other obstructing objects that are not shown in the building raster and would prevent trees from being planted in such locations.
The extent of the *Planting area* can be set to the same size as the extent of the output data from SOLWEIG. In this case, a buffer zone will be enforced to avoid edge effects. Another option in the tool will be to disable shading outside of the study area.

Optional settings can be set under *Advanced Parameters*. These include options to use either the random or the genetic starting algorithms, as well as number of iterations. Furthermore, an option to use a greedy algorithm, instead of the hill climbing





algorithm is included. The greedy algorithm is a faster option, which can be useful when requesting positions for a larger number of trees in extensive model domains. However, as shown in the results and mentioned in the discussion, tree locations can differ considerably between the hill climbing algorithm and the greedy algorithm.

The output from the TreePlanter is an updated raster CDSM and a vector point file with the positions of the new trees.

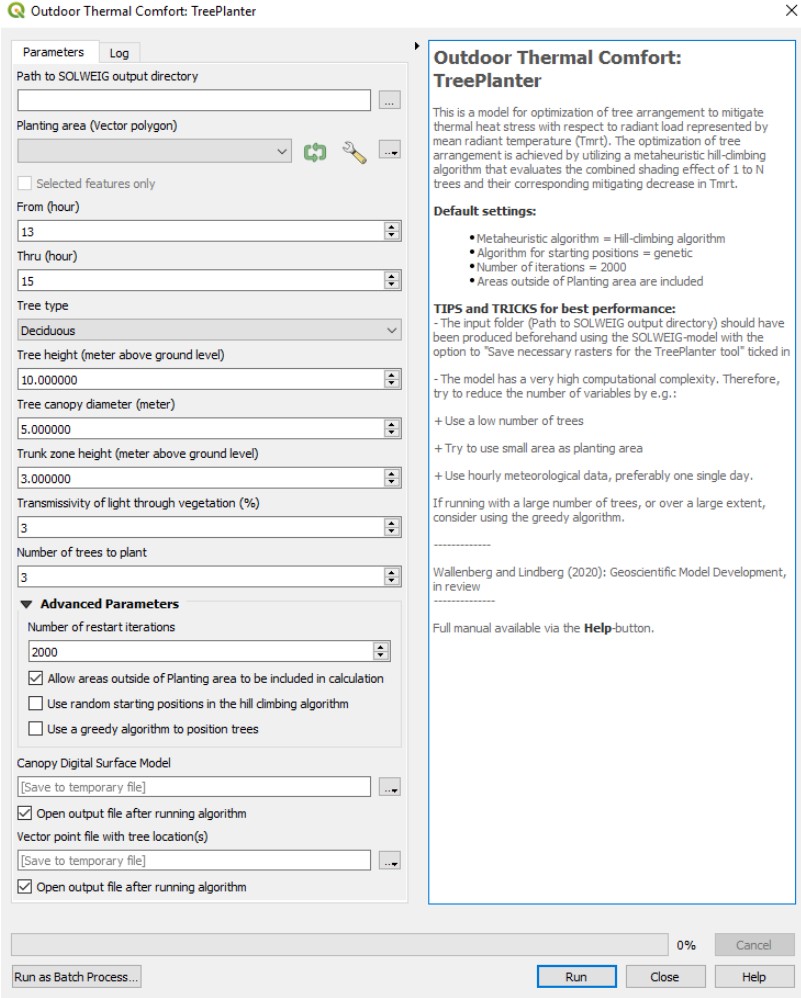

**Figure 12** Graphical user interface of the TreePlanter 1.0 in QGIS

## 6 Concluding remarks

The TreePlanter model presented in this paper have several advantages for future studies of mitigation of $T_{mrt}$ and analysis of shadow patterns in urban areas. Conclusions from the model performance analysis and the case study are:

- Modelling and optimization of positioning of trees with respect to mitigation of $T_{mrt}$ is very complex and
computationally extensive. TreePlanter and its metaheuristic hill climbing algorithm can give guidance in this issue.





- Both algorithms investigated (hill climbing and greedy) find tree locations that result in substantial decrease in $T_{mrt}$, and thus increase thermal comfort in exposed areas on clear, hot days. The locations, however, can differ considerably between the two algorithms.

- Tree locations depend primarily on tree size and time of day. Tree size indicates that juvenile and mature trees have different optimal positions, which can be important in e.g. urban planning.

- The hill climbing algorithm incorporates the combined shading effects of several trees, which are not addressed in the greedy algorithm.

- The model presented in this paper can give advice to urban planners and others in mitigating and improving thermal comfort in outdoor urban settings.

**Code and data availability**

Code is available from: http://doi.org/10.5281/zenodo.4616761

Test dataset is available from: http://doi.org/10.5281/zenodo.4616770

Tutorial on how to install code in QGIS from zip file, run model and description of model available from:

http://doi.org/10.5281/zenodo.4767387

**Author contributions**

NW led the development of TreePlanter and prepared the original draft. FL and DR provided overall supervision as well as review and editing of the original draft.

**Competing interests**

The authors declare that they have no conflict of interest.

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
