# Peer review of "Locating trees to mitigate outdoor radiant load of humans in urban areas using a metaheuristic hill climbing algorithm – Introducing TreePlanter v1.0"

_Geoscientific Model Development, 2021_

## Author Response (AR1)

The authors present a thorough description of their model – TreePlanter v1.0. Though the concept of modeling reduction in mean radiant temperature based on tree location is not entirely novel, their use of a metaheuristic approach speeds up computational time by avoiding a brute force approach, making their open source tool novel. Assumptions are clearly outlined and model limitations discussed.

**RC1:**

The paper would benefit from expanded quantitative evidence/discussion of how each of the various model output surfaces compares spatially. Though they offer discussion/values of the temperature reduction potential of various runs, there is no evidence offered for the spatial comparison of model output beyond qualitative discussion of visual qualities of the surfaces (throughout) and the mention that tree/shadow size affects placement (Section 5). Discussion/comparisons of model output would be further substantiated with the inclusion of quantitative comparisons of the spatial qualities of the reduction in mean radiant temperature. How much area is reduced by how much? Is total overall reduction what is meant by line 11 in Figure 2 (Pseudocode) or is it something else? It should be elucidated in the body of the text and in figures. Include summary statistics and discussion of global/zonal/focal operations to highlight how these comparisons are made in the model and by the modeler. Inclusion of these comparisons throughout would clarify the description of this tool and substantiate the conclusions in the paper.

AC1:

Thank you for your helpful comments! The inclusion of a table with summary of statistics proved to be quite beneficial as it provided us with some information which we were previously only speculating around. For example it made us realize that the greedy algorithm certainly is an excellent option. The results are complimentary to the hill climbing algorithm, as the resulting tree locations from this algorithm are quite different from the hill climbing algorithm, but the reduction in Tmrt, in this case the study area, is equal or sometimes even more. From an urban planning perspective these two algorithms can therefore work as alternatives to each other in that they potentially result in different tree locations. Furthermore, the quantification of results shows what we discussed in Sect 5.1 (Model limitation and potential), that recalculation of SVF (sky view factor) have a minor effect on cooling outside the shaded area, only contributing between 3.3-19.1 % of the total mitigation of average Tmrt (mean radiant temperature) in the entire raster. Here it is important to add that these 3.3-19.1 % are spread out over a large number of pixels (all pixels from which the new trees are visible). It should also be mentioned that even though the greedy algorithm sometimes has a higher reduction in Tmrt in the study area, the effect per shaded pixel is more efficient for tree locations established by the hill climbing algorithm. In addition, pseudocodes have been clarified and further elucidated in the text.

Changes in manuscript:

Lines 155-165 in relation to the pseudocode: "The model is run for *i* number of iterations as stated in line 4. Starting positions are determined in line 5 (see Sect. 2.2.2). The $\Delta T_{mrt}$ raster is then explored until all trees are in their local optimums. In line 11 total potential decrease in $T_{mrt}$ in the area shaded by the moving trees, integrated

over all time steps used, is estimated. The tree locations will continue to change as long as $T_{mrt}$ mitigation continues to increase in efficiency, as seen in lines 12-14. When there are no better locations for the trees and all are in local optimums the iteration stops, tree locations and corresponding decreases in $T_{mrt}$ are saved (line 15-17) and a new iteration commences (line 4). When the model has finished all iterations, it returns the locations of the trees from the iterations with largest decrease in $T_{mrt}$ (line 18)."

Added table with summary statistics comparing different surfaces and different model runs and text describing the results in lines 440-460. Changed and added text to discussion in relation to the summary statistics (lines 515-530).

**RC1:**

One additional minor point that may be worth expanding on is the discussion of model performance in real world settings v the study area (beginning on line 554) and throughout when "theoretically" is used to discuss alternative data and model parameters. Have any larger tests been run on real world scenarios? Have alternative data sources been tested? How does the model perform? Adding some discussion of tests might clear up ambiguities around the real-world application of this tool.

AC1:

Although we are interested in how the model performs in other real world settings than the study area, the scope of this paper is to demonstrate the model. Your comment is definitely appropriate and future work includes testing this on larger areas and different urban environments, with scenarios with many more trees. Ideally it would also be worth investigating if locations determined by the model are feasible or if underground aspects, e.g. pipes, root space, etc., would prevent tree planting. The greedy algorithm could be a great option for studies on larger number of trees especially if the area is large and possible locations as well as areas with excessive heat are large. In a small area with competition for space, on the other hand, the hill climbing algorithm and its functionality of moving trees simultaneously could be an advantage. Testing the two algorithms, and their pro's and con's, are certainly aspects of future work.

Changes in manuscript:

"A thorough evaluation of the algorithms included in the model as well as their pros and cons are aspects for future studies. How do their results compare? Under what circumstances are the different algorithms applicable?" added to lines 605-510.

Anonymous Referee #2

The authors have presented an interesting new tool TreePlanter that can be freely used in QGIS (as part of UMEP). Its main objective is to use trees to mitigate outdoor radiant load of humans in urban areas using a metaheuristic hill climbing algorithm by proposing the optimal location of trees in urban areas. Accordingly, TreePlanter can assist urban

planners in future research on optimization of tree planting in urban areas to increase

thermal comfort.

My main comments regarding some technicalities and unclear parts of the text are

provided in the attached document. Answering these comments can further improve and

clarify TreePlanter tool.

Please also note the supplement to this comment:

https://gmd.copernicus.org/preprints/gmd-2021-81/gmd-2021-81-RC2-supplement.pdf

AC1:

Thank you for your valuable comments! They have helped us clarify some parts which might have come out unclear. Specific answers to questions and comments are given below.

**RC2 Comment #1 @ line 130-135:**

Difference between random vs genetic?

AC1:

The differences between the random and genetic algorithms is how the starting positions a assigned, and is described in Sect 2.2.2 (typo in text fixed from "2.2.1" to "2.2.2").

**RC2 Comment #2 @ line 145-150:**

If you would use all of these settings with hill climbing (if possible), what would be the required time?

How do you define the necessary number of iterations? On what does that depend?

AC1:

The examples in the text are just to show the extensive number of brute-force calculations required to find optimal locations in comparison with example numbers with the hill climbing algorithm. The numbers in the hill climbing algorithm could be increased a lot and still not come close to the brute-force calculations.

The necessary number of iterations is set by the user, but would in the end depend on e.g. how large the model domain and "planting area" is, how many time steps are used, number of trees, size of trees, etc. Increasing the

number if iterations will increase model runtime, but also increase the probability of finding suitable locations for the trees.

Changes in manuscript:
Added "to the $2.5 * 10^{11}$ required with brute-force calculations" to end if line 150 for clarification and easier comparison with the example number with the hill climbing algorithm.

**RC2 Comment #3 @ line 170-175:**
What is defined by "close enough"? Depending on tree crown diameter or something else?

AC1:
Thank you for your comment. This, obviously, needs a more detailed explanation! "Close enough" means if it is even possible for the tree shadows to overlap. This is estimated by calculating the Euclidean distance between the longest parts of the tree shadows. Following clarification will be added to the revised manuscript: "Here, large distance is defined as the largest Euclidean distance between the northeast and southwest corner or the largest Euclidean distance between the northwest and southeast corner, of the estimated tree shadow for the generated tree."

**RC2 Comment #4 @ line 215-220:**
How do you decide this? On what does it depend?

AC1:
This is defined by the user (user of the tool/model) It can be set in the model GUI (*Number of trees to plant*).

Changes in manuscript:
"The user decides the number of trees ($k$) to optimize locations for in respect of $T_{mrt}$ mitigation." On lines 225-230.
Further clarification in text on line 230:
"Repeat from step 7 for $i$ number if iterations, with either the random or genetic starting algorithm, where $I$, the number of iterations, is set by the user."

**RC2 Comment #5 @ line 245-250:**
Add geographical coordinates and elevation.

AC1:
Coordinates have been added (approximately line 260).

**RC2 Comment #6 @ line 260-265:**
Why these two periods? Is the second part of the first period?

AC1:

The second period corresponds to a short time span in the afternoon when excessive heat is most pronounced.

Changes in manuscript:

Clarification in lines 275-285.

**RC2 Comment #7 @ line 260-265:**

Not clear.

AC1:

Tree sizes small, medium and large are described in Table 1.

Changes in manuscript:

Added "(see table Table 1 for description of tree sizes)" in line 275.

**RC2 Comment #8 @ line 265-270:**

Why this number?

AC1:

Indeed, this needs clarification. This is for the model to have enough iterations to avoid that the tree locations are in positions which are not necessarily optimal or satisfactory. Of course, we do not know if the locations determined after 20000 iterations are optimal, but they should definitely be adequate (as described in Section 4). To conclude, we chose this number because it is a very large number of iterations, representing the best-possible result from the algorithm.

Changes in manuscript:

"Running the model with 20000 to ensures that tree locations are satisfying, as is described in Sect. 4." In line 280.

**RC2 Comment #9 @ line 265-270:**

Why locations for trees differ substantially due to their size? This should be discussed here or in Discussion section.

AC1:

This is in Sect. 5 (Discussion). The main reason for the different positions with different tree sizes is that the trees would cast shadows of different sizes. The model is designed to use the tree shadows as efficiently as possible. This means that if two tree shadows overlap they only count once for the overlapping pixels. Therefore, the model would locate the trees in places where they do not overlap other tree shadows.

**RC2 Comment #10 @ line 285-290:**

What about individual hours? Are there big changes?

AC1:

Interesting question. We considered adding examples of individual hours, but eventually decided to exclude them as we find it likely that users will base model runs on several time steps. The reason for this is that heat stress not necessarily is confined to one time step. For example, high Tmrt (over 50-60 °C) is likely to occur over a longer time span on hot and clear days in e.g. Gothenburg. Still, to answer your question, tree locations will differ a lot if running with single time steps as an effect of the azimuth of the sun. A sun azimuth in the east will, for example, increase Tmrt in front of sunlit west-facing facades and a sun azimuth in the west will, similarly, increase Tmrt in front of sunlit east-facing facades.

**RC2 Comment #11 @ line 300-305:**

Edit language.

AC1:

Thank you! Clarified in line 320: "In the last example, using large tree size".

**RC2 Comment #12 @ line 340-345:**

Results from this table should be discussed in the main text. We can see that genetic starting algorithm is generally better. However, when 6+ trees are used, it is worse than random starting algorithm? Why this happens?

AC1:

Thank you for this important question! This is analyzed and discussed in Sect. 5 (Discussion). The way the genetic algorithm for starting positions works is that it will determine the starting positions of an iteration based on the "optimal" positions in the previous run. Therefore, as the model runs, the starting positions will come closer and closer to "warm" areas in the Tmrt rasters, which in this case are confined close to the walls of the buildings in the center of the "Planting area". This means that if the trees are starting in positions which are close to each other the model will have to adjust for overlapping shadows and adjusting for overlapping shadows is time consuming.

**RC2 Comment #13 @ line 340-345:**

Are these numbers in seconds (s)? If yes, add (s) in column names above them.

AC1:

Thank you for noticing! This has been fixed ((s) added to the column names in Table 2 where appropriate).

**RC2 Comment #14 @ line 350-355:**

Can you explain/discuss why this happens?

AC1:

This is mentioned in the discussion and could be explained by the fact that metaheuristics are not guaranteed to find optimal solutions. Nevertheless, the solutions found here (i.e. tree locations) have high mitigating potential. One way of lowering the chance of this is by increasing the number of iterations.

**RC2 Comment #15 @ line 365-370:**

Why this happens?

AC1:

This is discussed in Sect. 5 (Discussion).

**RC2 Comment #16 @ line 460-465:**

Language should be improved.

AC1:

Thank you! Sentence will be revised.

Changes in manuscript:

Previous sentence changed to "Thus, tree size can be an important factor when optimized mitigation of high radiant load is desired."

**RC2 Comment #17 @ line 585-590:**

Yes, this is very important and I am glad that authors mentioned this. I am wondering how many of the trees could be really planted in the study area if this was considered (rhetorical question). I guess that would be quite a small number (maybe even 0).

AC1:

Thank you for your comment! This is definitely something for future improvement of the model. It would, however, require data which is hard to acquire.

**RC2 Comment #18 @ line 585-590:**

Thank you! ☺

AC1:

You're welcome! ☺

**RC2 Comment #19 @ line 615-620:**

Yes, this tool can be an important first step in the decision-making process where to plant trees in cities. However, the situation on the ground would certainly modify the proposed solution to some extent.

AC1:

Yes, the situation underground (i.e. soil characteristics, root spacing, etc) would definitely modify the proposed solution. However, the tool, as you mention, can be used as a first step in decision-making. Future work should include an evaluation of these locations and their underground properties.

Changes to manuscript:

Added "Aspects of future work includes a comprehensive evaluation of the hill climbing and greedy algorithms. Furthermore, assessing the underground conditions is of importance. Model developments include running with different tree sizes simultaneously."

AC1:

Yes, the situation underground (i.e. soil characteristics, root spacing, etc) would definitely modify the proposed solution. However, the tool, as you mention, can be used as a first step in decision-making. Future work should include an evaluation of these locations and their underground properties.